# A Cross-Sectional Survey of Different Types of School Bullying before and during COVID-19 in Shantou City, China

**DOI:** 10.3390/ijerph20032103

**Published:** 2023-01-24

**Authors:** Linlin Xie, Qingchen Da, Jingyu Huang, Zhekuan Peng, Liping Li

**Affiliations:** 1School of Public Health, Shantou University, Shantou 515041, China; 2Injury Prevention Research Center, Shantou University Medical College, Shantou 515041, China; 3Center for Disease Control and Prevention of Guangzhou Huangpu District, Guangzhou 510799, China

**Keywords:** COVID-19, pandemic, bullying, school, victimization, perpetration, adolescent, risk behaviors

## Abstract

Background: Since the end of 2019, the COVID-19 pandemic has had serious wide-ranging effects on academic, occupational and other daily activities. Like other types of institutions, schools are facing unprecedented challenges. Students may face a variety of adverse consequences, including sleep disturbances and school bullying, if they are unable to adjust to the current learning and living environment. This study explored the effects of the COVID-19 pandemic on school bullying. Methods: A total of 5782 middle school students were enrolled in this multi-stage, cross-sectional study (3071 before and 2711 during the pandemic). The pre-pandemic group had a mean age of 14.9 ± 1.73, the pandemic group of 14.75 ± 1.47. Three models were set up using binary logistic regression to adjust for confounding variables (gender, school type, alcohol consumption, smoking, playing violent video games). Results: All types of bullying victimization and perpetration (physical, verbal, social and property bullying) were more common during the pandemic than before the pandemic. In terms of bullying victimization, property bullying victimization (crude odds ratio [OR]: 2.398, 95% CI: 2.014–2.854, *p* < 0.001; model 2 adjusted OR: 2.344, 95% CI: 1.966–2.795, *p* < 0.001; model 3 adjusted OR: 2.818, 95% CI: 2.292–3.464, *p* < 0.001) increased the most. In terms of bullying perpetration, verbal bullying perpetration (crude OR: 3.007, 95% CI: 2.448–3.693, *p* <0.001; model 2 adjusted OR: 2.954, 95% CI: 2.399–3.637, *p* < 0.001; model 3 adjusted OR:3.345, 95% CI: 2.703–4.139, *p* < 0.001) increased the most. Conclusion: This study corroborate the significance of the pandemic on traditional school bullying and suggests that we should further consider other types of bullying and establish and improve the response and prevention mechanisms during public health emergencies such as the COVID-19 pandemic.

## 1. Introduction

Several scholars have recognized that the COVID-19 pandemic has been particularly challenging for students at vulnerable stages of educational progress [1], with major consequences affecting students’ academic, extracurricular activities and other activities [2]. If students are not able to adapt well to the current learning and living environment, this could result in various physical and mental health problems, such as sleep disturbances, alcohol and substance misuse, and suicidal ideations or non-suicidal self-harm [3,4,5,6,7]. All of these effects may be cause by school bullying, and we focused on this aspect of the COVID-19 pandemic.

School bullying can be defined as pupils being bullied or victimized at school when they are consistently exposed to unfavorable acts by one or more other classmates [8], characterized by repeated victimization in a context of power imbalance [9]. School bullying has negatives effect on child and adolescent health [10]. Not only does it harm victims, but it also has adverse effects on the bullies [11]. For victims, bullying not only brings about bodily damage and pain, such as skin bruising, headaches and abdominal pain [12], but it also causes significant mental health problems, such as depression, suicidal and self-harming behavior, anxiety and social harm [13,14,15,16]. For bullies, bullying can be associated with alcohol and substance abuse, decreased performance, deteriorated health outcomes, earlier dating, negative emotions, violence and criminal behaviors, and other consequences [17,18,19,20]. Globally, bullying is recognized as a widespread public health problem [21]. One in three children have been bullied in the past 30 days, although there is substantial regional variation in the prevalence and type of bullying experienced [22].

Traditional bullying can be categorized as physical bullying (e.g., physical assault), verbal bullying (e.g., name-calling, insults and threats), social manipulation (e.g., exclusion and rejection), and property bullying (e.g., theft and vandalism) [23]. The commonest type of bullying is verbal bullying, followed by physical bullying [24]. Studies have shown that there are gender-specific differences in the prevalence of bullying [25]. Boys, as opposed to girls, are more likely to engage in physical and verbal bullying [26]. Girls are more prone to engaging in relational bullying and more prone to being victims of verbal and relational bullying [27]. The prevalence of bullying varies by grade level, with a peak in all types of bullying during middle school, particularly verbal bullying, which has been demonstrated to increase in prevalence from 32.6% in grade 5 to 78.5% in grade 8 [24]. Then the prevalence of bullying subsequently gradually decreases as the grade level increases [28]. From the ninth to twelfth grades, the prevalence of property victimization has been shown to drop significantly, from 21.4% to 10.6%, and overall bullying dropped by nearly half, from 32.5% to 17.8% [29]. Furthermore, studies have demonstrated that the prevalence of school bullying is associated with parental support levels, family background, friends’ assistance, academic performance, school attachment and other factors [30,31,32,33].

Given the enormous impact of the pandemic on human health, several scholars have conducted comparative studies on various aspects before and during the pandemic; however, there has been little comparative research on school bullying. Therefore, this study analyzed survey data on school bullying among secondary school students before and during the pandemic. This study aimed to understand whether the incidence of different types of school bullying differed between the pre-pandemic and pandemic periods. The study also aimed to determine whether the pandemic has had an impact on school bullying so as generate informative data to facilitate prevention and mitigation of school bullying.

**Hypothesis 1** **(H1).***The prevalence of traditional school bullying was different before and during the pandemic*.

## 2. Method

### 2.1. Participants

A total of 3097 middle school students in Shantou City in June 2018 were selected as the pre-pandemic group, along with a total of 2843 middle school students in Shantou City in June 2021 as the pandemic group. We recruited students in the seventh, eighth, tenth and eleventh grades; students in ninth and twelfth grades were not included to avoid distracting their preparations for standardized examinations. In the pre-pandemic group, the response rate was 99.2%, and 3071 questionnaires were collected. There were 1543 males and 1463 females, 1563 middle school students and 1508 high school students. In the pandemic period group, the response rate was 95.4%, and 2711 questionnaires were collected. There were 1465 males and 1246 females, 1553 middle school students and 1158 high school students.

### 2.2. Sample Size Calculation

The sample size was determined using the single population proportion formula (n = (Z _α_) ^2^ P (1 − P)/d^2^); whereas n = sample size, Z_α_ (1.96): significance level at α = 0.05, P: expected proportion (15%), d: margin of error = 0.1 P (0.015) and 20% non-response rate. According to that formula, sample size for this study should not be less than 2613 participants to be statistically acceptable, so 3071 participants before the pandemic and 2711 participants during the pandemic have been analyzed.

### 2.3. Procedure

We used a cross-sectional design with multi-stage stratified cluster sampling. The seven districts (counties) of Shantou City were divided into three parts: intermediate cities, fringe cities and island cities. Two junior high schools and two senior high schools were randomly selected in the intermediate cities, two junior high schools and two senior high schools were randomly selected in the peripheral cities, and one junior high school and one senior high school were randomly selected in the island cities. Baseline data (gender, age, only-child or not, school type, accommodation) and health risk behaviors (alcohol consumption, smoking, playing violent video games) were collected from all participants through questionnaires administered by trained researchers. All study participants provided written informed consent after being provided with detailed information about the study. A brief check was carried out when participants submitted the questionnaire. If any responses were omitted or inappropriate, participants were requested to re-submit the corresponding questionnaire items. After invalid questionnaires (e.g., those with incomplete, mismatched, missing, or unsuitable responses for at least a quarter of the questionnaire items) were excluded, the data were double-entered and crosschecked with EpiData 3.1 software.

### 2.4. Models

To adjust for confounding variables, three models were generated using binary logistic regression. Model 1 did not exclude any confounders and only time was included as a covariate. Based on model 1, model 2 excluded social and demographic characteristics (gender, school type). Finally, in model 3, health risk behaviors (alcohol consumption, smoking, playing violent video games) were excluded, based on model 2. Odds ratios (ORs) and 95% confidence intervals (CIs) were calculated before and after adjustment for the confounding factors.

### 2.5. Measurement

#### 2.5.1. Bullying Questionnaires

Bullying victimization and perpetration were measured using the Multidimensional Peer Victimization Scale (MPVS) and the Multidimensional Peer Bullying Scale (MPVS-RB), respectively [23,34]. The MPVS and MPVS-RB scales assess various kinds of bullying (physical, verbal, relational and property bullying) that happen during childhood and adolescence. At the outset of the investigation, candidates were presented with a formal definition of bullying [35] and instructed to review bullying victimization and perpetration during a specified period (within the semester). The two scales are divided into four dimensions and 16 items, with each item being scored according to a 5-point Likert scale [36], ranging from 0 (always) to 4 (never). The scores of the four dimensions range from 0 to 16. In the MPVS scale, physical bullying victimization items included: “Someone punched me”, “someone kicked me”, etc.; verbal bullying victimization items included: “Someone gave me a nickname”, “someone made fun of me because of my appearance”, etc.; relational bullying victimization items included: “Someone tried to turn my friends against me and distance me”, “someone made people not talk to me”, etc.; property bullying victimization items include: “Someone took my stuff without my permission”, “someone tried to destroy my stuff”, etc. In the MPVS-RB scale, physical bullying items included: “I punched someone”, “I kicked someone”, etc.; Verbal bullying included: “I gave someone a nickname”; “I made fun of someone because of his/her appearance”, etc.; relational bullying items included: “I tried to turn a classmate’s friends against him/her or distance him/her”, “I asked others not to talk to him/her”, etc.; property bullying items included: “I took other people’s belongings without their permission” and “I tried to destroy other people’s belongings”, etc. In the pre-pandemic survey, the Cronbach’s α coefficients for the four dimensions of the MPVS scale were 0.810, 0.881, 0.785 and 0.773. The Cronbach’s α coefficients for the four dimensions of the MPVS-RB scale were 0.882, 0.845, 0.814 and 0.889. In the pandemic survey, the Cronbach’s α coefficients for the four dimensions of the MPVS scale were 0.767, 0.741, 0.832 and 0.792. The Cronbach’s alpha values for the four dimensions of the MPVS-RB scale were 0.743, 0.834, 0.631 and 0.786. The Cronbach’s α coefficients of the questionnaire were all greater than or equal to 0.6, which indicated the stability of the questionnaire. If the score for a dimension is higher than 1, it demonstrates that the student is being bullied or bullying others in that dimension. If the overall score for the four dimensions is higher than 1, the student is being bullied or bullying others at school.

#### 2.5.2. Risk Behaviors Questionnaires

Questions include “How often do you smoke?”; “How often do you drink”; “How often do you play violent video game?”; The frequencies of health risk behaviors were measured on a 5-point Likert scale. Tobacco and drinking consumption ranged from 0 (never) to 4 (≥5 times/month). The frequency of violent video game plays ranged from 0 (never) to 4 (≥5 h/week). When the score is greater than 1, it indicates the occurrence of the health risk behavior.

### 2.6. Statistical Analysis

Statistical analysis was conducted using SPSS Statistics for Windows, version 26.0 (IBM Corp., Armonk, NY, USA) and GraphPad Prism 9. We analyzed the participants’ basic characteristics using descriptive statistics to determine the number and rates of bullying victimization and bullying perpetration in different variables (gender, school type, drinking, smoking, playing violent video games) before and during the pandemic. Binary logistic regression permitted adjustments for potential confounders. Missing values were accounted for using multiple imputation. Statistical testing was two-sided, at a significance level of α = 0.05.

## 3. Results

Table 1 shows the prevalence of overall bullying victimization and perpetration with different variables before and during the pandemic. A total of 3071 people participated in the survey before the pandemic, with 452 cases of bullying victimization and 186 cases of bullying perpetration. A total of 2711 people participated in the survey during the pandemic, with 768 cases of bullying victimization and 424 cases of bullying perpetration. As can be seen, the prevalence of both overall bullying victimization and perpetration increased during the pandemic, regardless of the variable.

Figure 1 shows the crude and model adjusted ORs for overall bullying victimization and overall bullying perpetration for before and during the pandemic. There was an increase in the incidence of overall bullying victimization during the pandemic (n = 4436) (crude OR: 2.290, 95% Cl: 2.011–2.609, *p* < 0.001; model 2 adjusted OR: 2.25, 95% CI: 1.973–2.566, *p* < 0.001; model 3 adjusted OR: 2.621, 95% CI: 2.235–3.074, *p* < 0.001). Similarly, the prevalence of overall bullying perpetration increased during the pandemic (crude OR: 2.876, 95% CI: 2. 400–3.446, *p* < 0.001; model 2 adjusted OR: 2.829, 95% CI: 2.356–3.398, *p* < 0.001; model 3 adjusted OR: 3.192, 95% CI 2.645–3.853, *p* < 0.001).

Figure 2 shows the crude and each model’s adjusted ORs for various kinds of bullying victimization (physical victimization, social manipulation, verbal victimization, property attacks) before and during the pandemic. The rates of all bullying victimization were higher during the pandemic than before the pandemic. The crude and model adjusted ORs for physical victimization, social manipulation, verbal victimization and property attacks were, respectively, as follows: crude OR: 1.698, 95% CI: 1.263–2.283, *p* < 0.001; model 2 adjusted OR: 1.665, 95% CI: 1.237–2.241, *p* < 0.001; model 3 adjusted OR: 1.788, 95% CI: 1.324–2.413, *p* < 0.001; crude OR: 1.688, 95% CI: 1.343–2.12, *p* < 0.001; model 2 adjusted OR: 1.641, 95% CI: 1.305–2.063, *p* < 0.001; model 3 adjusted OR: 1.693, 95% CI: 1.344–2.133, *p* < 0.001; crude OR: 2.361, 95% CI: 2.038–2.733, *p* < 0.001; model 2 adjusted OR: 2.307, 95% CI: 1.989–2.676, *p* < 0.001; model 3 adjusted OR: 2.417, 95% CI: 2.074–2.817, *p* < 0.001; crude OR: 2.398, 95% CI: 2.014–2.854, *p* < 0.001; model 2 adjusted OR: 2.344, 95% CI: 1.966–2.795, *p* < 0.001; model 3 adjusted OR: 2.818, 95% CI: 2.292–3.464, *p* < 0.001. The incidence of property attacks increased the most.

Figure 3 shows the crude and model adjusted ORs for different types of bullying perpetration (physical bullying, social manipulation, verbal bullying and property attacks) before and during the pandemic. The rates of all types of bullying were higher during the pandemic than before the pandemic. The crude and model adjusted ORs for physical bullying, social manipulation, verbal bullying and property attacks were, respectively, as follows: crude OR: 2.209, 95% CI: 1.487–3.280, *p* < 0.001; model 2 adjusted OR:2.143, 95% CI: 1.441–3.188, *p* < 0.001; model 3 adjusted OR: 2.201, 95% CI: 1.474–3.286, *p* < 0.001; crude OR: 2.752, 95% CI: 1.988–3.812, *p* < 0.001; model 2 adjusted OR: 2.65, 95% CI: 1.912–3.673, *p* < 0.001; model 3 adjusted OR: 2.972,95% CI: 2.123–4.161, *p* < 0.001; crude OR: 3.007, 95% CI: 2.448–3.693, *p* < 0.001; model 2 adjusted OR: 2.954, 95% CI: 2.399–3.637, *p* < 0.001; model 3 adjusted OR:3.345, 95% CI: 2.703–4.139, *p* < 0.001; crude OR: 2.013, 95% CI: 1.309–3.096, *p* < 0.001; model 2 adjusted OR: 1.955, 95% CI: 1.270–3.011, *p* < 0.002; model 3 adjusted OR: 1.936, 95% CI: 1.255–2.986, *p* < 0.003. The greatest increase was in the incidence of verbal bullying.

## 4. Discussion

### 4.1. Increases in the Incidence of Bullying Victimization and Perpetration during the Pandemic

The results of this study showed that the prevalence of bullying victimization and bullying perpetration was higher during the pandemic than before the pandemic, which was contrary to the findings of Vaillancourt et al. Their research observed higher rates of victimization and perpetration in the physical, verbal, and social bullying domains before the pandemic than during the pandemic period [37]. As for why the prevalence of school bullying declined during the pandemic, they explained that it was due to decreased class sizes, increased supervision and fewer opportunities for social interaction in schools. However, in Shantou City, China, no significant reduction in class sizes was observed and, due to the adoption of closed management, learning and social activities have increased. The increased prevalence of school bullying during the pandemic in this study may have been related to the expansion of the local COVID-19 outbreak and the prevention and control measures taken in schools [38]. There were few outbreaks in Shantou City, where this study was conducted, with a relatively small impact on the local population [39]. However, for the safety of the students, schools had adopted tightened infection prevention and control [2]. Students were not allowed to leave school at will, only for serious reasons, and students were required to undergo frequent nucleic acid tests. In order to ease students’ rebellious mood, the school organized more learning and social activities such as learning lectures and fun sports games. The measures taken by the school are closely related to the national infection prevention and control policy. The policies adopted in different regions also play an important role in the results. In 2021, Chinese epidemic prevention and control policies were based on the general strategies of “defending externally against importation, defending internally against rebound” and “dynamic zero-case” policy [40]. Therefore, mental health problems (such as anxiety and depression) are likely to have arisen as direct results of, for example, blocked access, frequent nucleic acid testing and fewer outings [41,42]. Several studies have shown COVID-19 related anxiety to be associated with psychiatric health problems among adults with uncommon disorders and that the prevalence of anxiety has been higher during the pandemic than before the pandemic [43]. Therefore, psychological problems arising from the COVID-19 pandemic (and the mitigatory responses thereto) are likely to have caused the increase in school bullying. This could be explained by the person–environment fit theory, which describes the relationship between the development needs of adolescents and changing circumstances in their lives [44]. These students were at a vulnerable stage and were exposed to a negative environment, undoubtedly increasing the probability of health problems [45].

In the setting of the COVID-19 outbreak in China, the incidence of depression and worsening symptoms among adolescents has increased [46]. There were several mechanisms linking depression and criminal behavior [47]. Adolescents suffering from major depression might opt to indulge in theft as a mechanism of defense or as a helpful way of receiving concern [48], but whether this indirect association is responsible for the increased incidence of property bullying during the pandemic remains to be investigated. Adolescents can be impulsive and relatively easily distracted by negative information [49]. Therefore, they have been particularly affected by the pandemic and this has yielded outward manifestations of negative emotions. When communicating with others, adolescents have difficulties controlling their emotions, which leads to more conflicts. As a result of blocked access, schools adopt more learning and social activities, which is likely to be causing an increase in the prevalence of physical and relational bullying and verbal bullying. The increases in activities give students more opportunities to communicate with each other, but if communication is inappropriate, verbal conflicts that intensify without reasonable intervention are likely to result in fights, isolation, insults, etc.

### 4.2. Strengths

A strength of this study lies in the fact that few scholars have studied the changes in school bullying in China arising from the pandemic, and in this study variables other than time were investigated to determine how the pandemic has impacted school bullying. Moreover, multi-stage, stratified cluster sampling was used to enroll students in this study, which optimized its representativeness of the whole population.

### 4.3. Limitations

Some limitations also existed in this research. Firstly, we employed a cross-sectional study design, making it difficult to establish causality. Therefore, future research on this topic could benefit from longitudinal studies to enhance causal inference. Secondly, the fact that participants before and during the pandemic were not the same population, as well as that the status of participants may have changed during the relatively long interval between surveys, may have affected the results of the study. However, the survey subjects were all secondary school students in the same area, while this paper set up three models to adjust for confounding variables, which can reduce bias to some extent. Thirdly, the impact of the COVID-19 pandemic (and response) on cyberbullying was not investigated in the study. With the advancement of technology, cyberbullying is becoming more and more common in our society [50]. The pandemic has compelled many schools and colleges around the world to turn to online education, resulting in increased online time for students, which may contribute to the growth in cyberbullying [51]. The impact of pandemics on cyberbullying has already been studied in several countries [52] and future research in this area should be carried out in China. Fourthly, the study did not consider bullying inside households. Sabah et al. observed that the effect of sibling physical and sibling verbal victims on school victimization was statistically significant [53]. Future studies could therefore consider bullying within the family to explore the extent to which the increase in the prevalence of all types of school bullying during the pandemic was related to school bullying and the extent to which it was related to bullying inside households. Finally, the results from this study were based on self-reported data, which is associated with some disadvantages. First, they might be susceptible to self-presentation strategies or social desirability bias [54]. Additionally, some recall bias may exist, as participants were required to recall events that had happened during the semester.

## 5. Conclusions

In general, the results of this study corroborate the significance of the pandemic’s effect on traditional school bullying. The pandemic was associated with increased rates of all types of traditional bullying victimization and perpetration investigated (physical, social, verbal, property bullying). This suggests that we should further consider other types of bullying and establish and improve the response and prevention mechanisms for public health emergencies such as the COVID-19 pandemic. The results of this study showed that the recurrence of epidemics and frequent emergencies such as blocked access might impact on increase of school bullying among adolescents. As a bridge between schools and students, teachers need to help students adapt to current school life and pay attention to their mental health issues. Reasonable coping mechanisms can reduce the harm of school bullying and play an effective role in preventing school bullying, ensuring the personal safety of all teachers and students on campus and creating a safe school environment.

## Figures and Tables

**Figure 1 ijerph-20-02103-f001:**
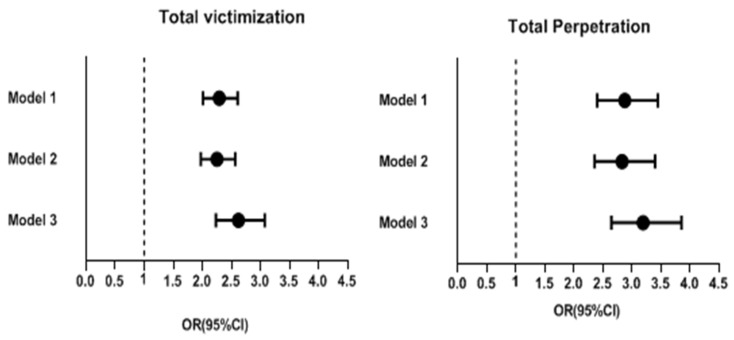
Crude ORs and adjusted odds ratios for the incidence of overall bullying victimization and perpetration before and during the pandemic (error bars represent 95% confidence intervals).

**Figure 2 ijerph-20-02103-f002:**
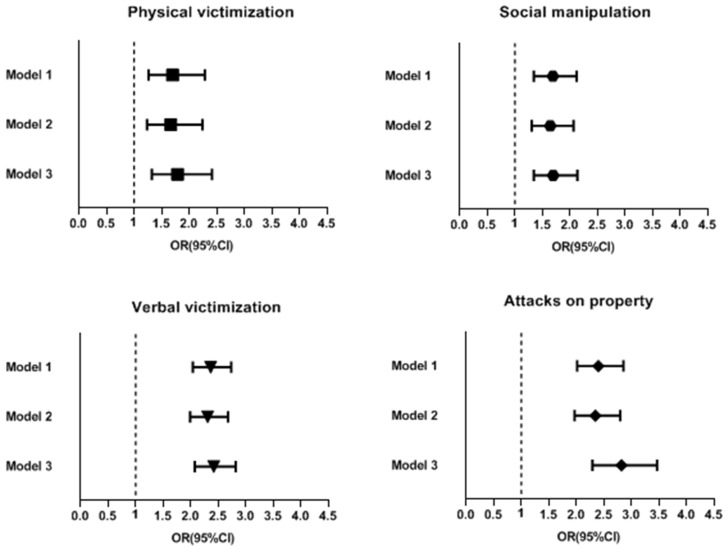
Crude ORs and adjusted odds ratios for the incidence of various kinds of bullying victimization before and during the pandemic (error bars represent 95% confidence intervals).

**Figure 3 ijerph-20-02103-f003:**
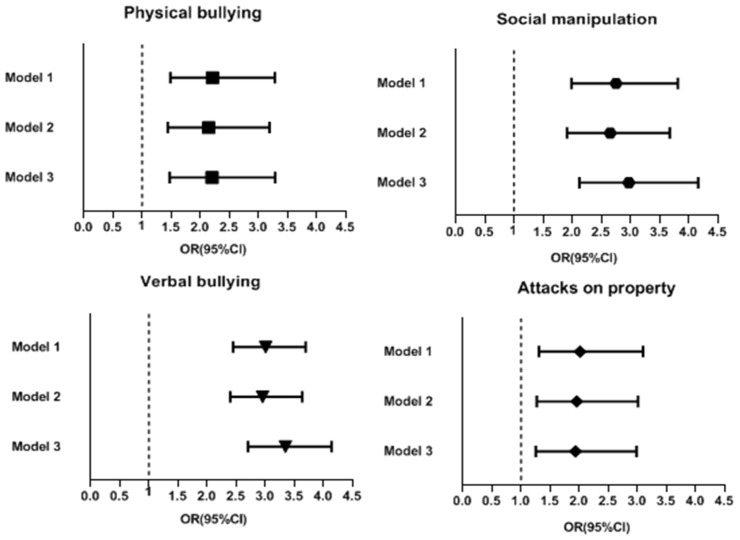
Crude ORs and adjusted odds ratios for the incidence of various kinds of bullying perpetration before and during the pandemic (error bars represent 95% confidence intervals).

**Table 1 ijerph-20-02103-t001:** Comparison of the prevalence of overall bullying victimization and bullying perpetration by different variables before and during the pandemic.

Variable *	Before COVID-19		During COVID-19	
	Bullying Victimization *n* (%)	Bullying Perpetration *n* (%)	Bullying Victimization *n* (%)	Bullying Perpetration *n* (%)
Gender				
Male	285 (18.47)	146 (9.46)	509 (34.74)	313 (21.37)
Female	167 (11.41)	40 (2.73)	259 (20.79)	111 (8.91)
School				
Junior High School	255 (16.31)	104 (6.65)	470 (30.26)	260 (16.74)
High School	197 (13.06)	82 (5.44)	298 (25.73)	164 (14.16)
Drinking				
No	405 (14.37)	140 (4.97)	672 (26.98)	356 (14.29)
Yes	47 (20.26)	46 (19.83)	96 (43.64)	68 (30.91)
Smoking				
No	427 (14.51)	162 (5.51)	733 (27.79)	393 (14.90)
Yes	25 (23.15)	24 (22.22)	35 (47.95)	31 (42.47)
Playing violent video games				
No	160 (10.61)	45 (2.98)	369 (22.12)	158 (9.47)
Yes	292 (19.52)	141 (9.43)	399 (37.89)	266 (25.26)

* There were 1.12% missing values for gender; 0.35% missing values for drinking; 0.36% missing values for smoking; 1.16% missing values for playing violent video games.

## Data Availability

The datasets used and analyzed in the study are available from the corresponding author on reasonable request.

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
