# Peer review of "A Cross-Sectional Survey of Different Types of School Bullying before and during COVID-19 in Shantou City, China"

_ijerph, 2023, doi:10.3390/ijerph20032103_

Round 1

Reviewer 1 Report

1. This is interesting study considering the differences in prevalence of different types of school bullying before and during COVID-19 pandemic. However the title does not exactly refer to the study. I suggest to rewrite it to define more specific what study is about.  

2. I suggest to correct English style  (line 18: property bullying victimization, line 21: verbal bullying perpetration; line 37: may be caused by school bullying; line 36: self-harm)

3. It was difficult to read the text with single references in the middle of sentences. Please combine them and refer in the end of the sentences or paragraphs according to the IJERPH/MDPI instructions for authors. Moreover there is a mess in references in the Introduction part. Please check them carefully,  find appropriate references and correct. The following references are not applicable: 22, 23,24,28,29,31,36. Please find the right ones and refer to appropriate studies.

4. In Abstract please include the mean age of the study sample in 2018 and 2021. In conclusions there is a lack of recommendations what kind of further study should be conducted to explain and prevent school bullying after COVID-19 pandemic.

5. Line 45-46: Depression, suicidal, self-harming behaviors, anxiety are mental health problems

6. Reg. line 75 this study aimed to understand longitudinal changes in different types of school bullying, among Chinese teens. Tthere is no information if researchers conducted observations of the same subjects (teens) over a period of time, so it means the same participants in period of 2018-2021.

7. Please explain the sentence in line 93-94 “If any responses were omitted or inappropriate, participants were requested to do the corresponding questionnaire items”. Moreover please provide the response rate.

8. In 2.4.1. Please provide the reference to Multidimensional Peer Victimization Scale (MPVS) and the Multidimensional Peer Bullying Scale (MPVS-RB), and describe the process of adaptation and validation of these tools to Chinese students population. Please provide the reliability of applied scales by Cronbach’s coefficients, as it is written in line 114: The two scales are divided into four dimensions and 16 items.

9. In 2.4.2. Please describe the measures of each risk behavior by specific scales, and how they were scored and calculated.

10. In 2.5. Please justify and describe conducted statistical analysis,  both descriptive and regression analysis. A significance level should be set up at p<0.05 not “of 0.05”.

11. Results in Table 1 are not clear. If this is longitudinla study N of study sample should be 2711 (the same participants before and during the pandemic). If not please clarify. Moreover please provide results of differences in victimization in 2018 and 2021 , and between perpetration in 2018 and 2021, and provide missing values in %. Please show the results of correlations matrix to describe the relationships between variables. Please correct title of Fig. 1, since model 1 shows crude ORs.  Figure 1 should be supported by supplementary materials with results of the regressions. Please provide goodness of fit of the presented models.

Regarding my above comments and remarks to introduction, measures, statistical analysis, and results it is not possible to comment further parts of the manuscript. I encourage Authors to rewrite, correct and complete the manuscript so that it could be fully peer-reviewed.

Author Response

Dear editor,

Thank you for editor’ and reviewers’ opinions. My article number is ijerph-2131360. These comments are very helpful to improve the quality of the manuscript. Now I response the reviewers’ comments with a point by point and highlight the changes in revised manuscript. Full details of files are listed. Thank you again for your valuable comments.

Yours,

sincerely

  1. This is interesting study considering the differences in prevalence of different types of school bullying before and during COVID-19 pandemic. However the title does not exactly refer to the study. I suggest to rewrite it to define more specific what study is about.

Response: We appreciate it very much for this good suggestion, and we have done it according to your ideas. I have change the title to “A cross-sectional survey of different types of traditional school bullying before and during COVID-19 in Shantou City, China”.

  1. I suggest to correct English style(line 18: property bullying victimization, line 21: verbal bullying perpetration; line 37: may be caused by school bullying; line 36: self-harm)

Response: We appreciate it very much for this good suggestion, and we have done it according to your ideas. We are very sorry for our careless mistake and it was rectified at Line 20, Line 22 and Line 37.

  1. It was difficult to read the text with single references in the middle of sentences. Please combine them and refer in the end of the sentences or paragraphs according to the IJERPH/MDPI instructions for authors. Moreover there is a mess in references in the Introduction part. Please check them carefully,find appropriate references and correct. The following references are not applicable: 22, 23,24,28,29,31,36. Please find the right ones and refer to appropriate studies.

Response: We appreciate it very much for this good suggestion, and we have done it according to your ideas. I have merged the literature citations for better reading. At the same time, the inapplicable literature citations are replaced.

  1. In Abstract please include the mean age of the study sample in 2018 and 2021. In conclusions there is a lack of recommendations what kind of further study should be conducted to explain and prevent school bullying after COVID-19 pandemic.

Response: We appreciate it very much for this good suggestion, and we have done it according to your ideas. The mean age has been added to the summary section at Line 17 and the conclusions  section has been modified at Line 26.

  1. Line 45-46: Depression, suicidal, self-harming behaviors, anxiety are mental health problems.

Response: We appreciate it very much for this good suggestion, and we have done it according to your ideas. We are very sorry for our careless mistake and it was rectified at Line 45.

  1. line 75 this study aimed to understand longitudinal changes in different types of school bullying, among Chinese teens.Tthere is no information if researchers conducted observations of the same subjects (teens) over a period of time, so it means the same participants in period of 2018-2021.

Response: We appreciate it very much for this good suggestion, and we have done it according to your ideas. I'm sorry for my our careless mistake. It was different middle school students before the pandemic and during the pandemic, but they were all middle school students from the same area. I have revised at Line 74-76. And it was also mentioned in the limitation that it was not a longitudinal study.

  1. Please explain the sentence in line 93-94 “If any responses were omitted or inappropriate, participants were requested to do the corresponding questionnaire items”. Moreover please provide the response rate.

Response: We appreciate it very much for this good suggestion, and we have done it according to your ideas. A brief check will be carried out when participants submit the questionnaire.If any responses were omitted or inappropriate, participants were requested to do the corresponding questionnaire items. We have added a supplement for the reader's better understanding at Line 109-110. The response rate was also added at Line 88 and Line 90.

  1. In 2.4.1. Please provide the reference to Multidimensional Peer Victimization Scale (MPVS) and the Multidimensional Peer Bullying Scale (MPVS-RB), and describe the process of adaptation and validation of these tools to Chinese students population. Please provide the reliability of applied scales by Cronbach’s coefficients, as it is written in line 114: The two scales are divided into four dimensions and 16 items.

Response: We appreciate it very much for this good suggestion, and we have done it according to your ideas. Few studies have validated MPVS and MPVS-RB questionnaires on Chinese middle school students. But many Chinese studies of bullying use the scale.e.g.Li, L., Chen, X., & Li, H. (2020). Bullying victimization, school belonging, academic engagement and achievement in adolescents in rural China: A serial mediation model. Children and youth services review113, 104946. Moreover, the results of two surveys in this study both showed a high Cronbach’s coefficient. It has been added to the literature at Line 149-156.

  1. In 2.4.2. Please describe the measures of each risk behavior by specific scales, and how they were scored and calculated.

Response: We appreciate it very much for this good suggestion, and we have done it according to your ideas. The description and calculation criteria of the risk factor scale have been supplemented at Line 161-162 and Line 165-166.

  1. In 2.5. Please justify and describe conducted statistical analysis,both descriptive and regression analysis. A significance level should be set up at p<0.05 not “of 0.05”.

Response: We appreciate it very much for this good suggestion, and we have done it according to your ideas. We are very sorry for our careless mistake and it was rectified at Line 172.

  1. Results in Table 1 are not clear.If this is longitudinla study N of study sample should be 2711 (the same participants before and during the pandemic). If not please clarify. Moreover please provide results of differences in victimization in 2018 and 2021 , and between perpetration in 2018 and 2021, and provide missing values in %. Please show the results of correlations matrix to describe the relationships between variables. Please correct title of Fig. 1, since model 1 shows crude ORs.  Figure 1 should be supported by supplementary materials with results of the regressions. Please provide goodness of fit of the presented models.

Response: We appreciate it very much for this good suggestion, and we have done it according to your ideas. The pre-pandemic and post-pandemic participants were different, but they were all secondary school students in the same area. The study sample was the sum of the pre-pandemic and post-pandemic sample sizes. We have revised it. Missing value have been changed to percentages at Line 183-184. At the same time, Figure 1 shows the number of overall bullying victimization and bullying perpetration with different demographic characteristics. We have revised the previous title due to ambiguity caused by insufficient clarity. The headings of Figures 1, 2, and 3 have also been revised. The degree of fit is rarely mentioned when using this model in literature reading, so it has not been calculated in this study. Secondly, this study mainly discussed the impact of the COVID-19 pandemic on the incidence of bullying. Factors other than the COVID-19 pandemic were taken as confounding factors, and other influencing factors other than the pandemic were not discussed in this study. Thank you for your comments on my manuscript. We benefited a lot from it. At the same time, the above are also my thoughts for you to ask questions, which we think can better let you understand the content of my article. We wish you success in work and good luck!

Reviewer 2 Report

The study addresses a relevant topic in the field. Although there are some comments to be made:

- Instruments. In order to have a better understanding of the different dimensions of bullying measured by the MPVS and MPVS-RB scales, some items should be included as examples of each dimension. Also, it should be included the authors of each of the scales in the references.

- Participants. The characteristics of the sample of participants need to be more clearly described: the number f participants by age, gender, grade, type of school, socio-economic status of schools, etc. It would be good to include a table of participants showing this sociodemographic information. 

Also, if one of the objectives of this study is to know about the different types of school bullying in China before and during the Covid, more specific context of how schools in China were working during the pandemic should be included, or the program of learning students were taking. This could contribute to give a more accurate picture of the type of communication between peers from school that took place online during the time of remote school, if this was the case.

- Results. The results in Table 1 should be described more clearly. Data are not clear, for example, it is not clear how bullying victimization and bullying perpetration data were measured for each of the different variables. Besides, data referring to the different demographic characteristics are not clear either, considering that we do not have the overall data to be able to interpret the corresponding results and percentages.  Specifically, it should be stated how the data were calculated from the general questions into binary bullying experience groups for bully victimization and bully perpetration.

The same happens with the data on risky behaviors. It is not clear how they are measured, considering that in Table 1 they are treated as dichotomous variables (yes/no), while in the method section it is stated that these risk behaviors (drinking, smoking, playing violent video games) were measured using a likert scale. 

Likewise, references to previous studies in which statistically significant relationships have been found between bullying behaviors and the risk behaviors measured in this study (drinking, smoking, playing violent video games) should be included. In the introduction, there is also no reference to the existing debate as to whether or not violent video games contribute to social aggression and violence.

- Conclusions. With respect the conclusions, there are some important points to be made. The authors say that one of the objectives of this study is to determine how the pandemic has impacted school bullying. From my point of view there is an important problem with this statement, if we consider that in order to do that, this study should have addressed also the study of cyberbullying during the lockdown and remote schooling. To what extent is peer bullying being measured if cyberbullying that may have occurred in confinement contexts during the pandemic is not addressed? 

Another important point is that the lack of explanation of some of the main results obtained or a lack of reference to other studies conducted in this field that obtained opposed results with respect the increase of all types of bullying during the pandemic than before the pandemic. How do authors explain these contradictory results found in China with respect other countries? Referring to the fact of all types of bullying were higher during the pandemic than before the pandemic, but without giving any plausible explanation for these facts? And what about cyberbullying? Why cyberbullying was not approached to determine how the pandemic impacted school bullying? Or to what extent these results about the increase of all types of bullying during the pandemic could not be related to peer bullying or school bullying but instead with bullying inside the households during the pandemic? If this is so, to what extent these could be due to an important methodological problem that was not controlled or that escaped out of control of researchers? 

Author Response

Dear reviewer,

Thank you for editor’ and reviewers’ opinions. My article number is ijerph-2131360. These comments are very helpful to improve the quality of the manuscript. Now I response the reviewers’ comments with a point by point and highlight the changes in revised manuscript. Full details of files are listed. Thank you again for your valuable comments.

Yours,

sincerely

  1. In order to have a better understanding of the different dimensions of bullying measured by the MPVS and MPVS-RB scales, some items should be included as examples of each dimension. Also, it should be included the authors of each of the scales in the references.

Response: We appreciate it very much for this good suggestion, and we have done it according to your ideas. We have added the specific items of each scale at Line 136-148 and quoted the references of each scale.

  1. The characteristics of the sample of participants need to be more clearly described: the number f participants by age, gender, grade, type of school, socio-economic status of schools, etc. It would be good to include a table of participants showing this sociodemographic information. 

Response: We appreciate it very much for this good suggestion, and we have done it according to your ideas. The relevant information has been added to the participants part at Line 87-92. However, due to the lack of detail in the questionnaire, the specific types of schools and economic conditions could not be added.

  1. Also, if one of the objectives of this study is to know about the different types of school bullying in China before and during the Covid, more specific context of how schools in China were working during the pandemic should be included, or the program of learning students were taking. This could contribute to give a more accurate picture of the type of communication between peers from school that took place online during the time of remote school, if this was the case.

Response: We appreciate it very much for this good suggestion, and we have done it according to your ideas. In order to give readers a better understanding of the situation of middle schools in China during the pandemic, we added to the discussion at Line 245-254.

  1. The results in Table 1 should be described more clearly. Data are not clear, for example, it is not clear how bullying victimization and bullying perpetration data were measured for each of the different variables. Besides, data referring to the different demographic characteristics are not clear either, considering that we do not have the overall data to be able to interpret the corresponding results and percentages.  Specifically, it should be stated how the data were calculated from the general questions into binary bullying experience groups for bully victimization and bully perpetration.

Response: We appreciate it very much for this good suggestion, and we have done it according to your ideas. The title and data for Table 1 have been revised. Table 1 shows the prevalence of school overall bullying victimization and perpetration among secondary school students with different demographic characteristics before and during the pandemic. The first grid shows the number and proportion of male students who were bullied before the pandemic. And the specific criteria is " If the overall score for the four dimensions is higher than 1, the student is being bullied or bullying others at school.”

  1. The same happens with the data on risky behaviors. It is not clear how they are measured, considering that in Table 1 they are treated as dichotomous variables (yes/no), while in the method section it is stated that these risk behaviors (drinking, smoking, playing violent video games) were measured using a likert scale. 

Response: We appreciate it very much for this good suggestion, and we have done it according to your ideas. The items and measurement methods for risk factor measurement have been supplemented at Line 161-162 and Line 165-166.

  1. Likewise, references to previous studies in which statistically significant relationships have been found between bullying behaviors and the risk behaviors measured in this study (drinking, smoking, playing violent video games) should be included. In the introduction, there is also no reference to the existing debate as to whether or not violent video games contribute to social aggression and violence.

Response: We appreciate it very much for this good suggestion. You're absolutely right that bullying is related to these risk factors, but we have a few thoughts on this suggestion. As these risk factors are considered confounding factors in this paper, the impact of the pandemic on the incidence of school bullying is mainly discussed, so these risk factors are not discussed in detail.

  1. With respect the conclusions, there are some important points to be made. The authors say that one of the objectives of this study is to determine how the pandemic has impacted school bullying. From my point of view there is an important problem with this statement, if we consider that in order to do that, this study should have addressed also the study of cyberbullying during the lockdown and remote schooling. To what extent is peer bullying being measured if cyberbullying that may have occurred in confinement contexts during the pandemic is not addressed? 

Response: We appreciate it very much for this good suggestion, and we have done it according to your ideas.Yes, what you said is also one of the limitations of this paper, because the previous manuscript has been revised to the traditional school bullying to clarify the research topic. Cyberbullying is not included in traditional bullying, but it is one of the limitations of this study that we will consider in future studies and that we also mentioned in the discussion of limitations.

  1. Another important point is that the lack of explanation of some of the main results obtained or a lack of reference to other studies conducted in this field that obtained opposed results with respect the increase of all types of bullying during the pandemic than before the pandemic. How do authors explain these contradictory results found in China with respect other countries? Referring to the fact of all types of bullying were higher during the pandemic than before the pandemic, but without giving any plausible explanation for these facts? And what about cyberbullying? Why cyberbullying was not approached to determine how the pandemic impacted school bullying? Or to what extent these results about the increase of all types of bullying during the pandemic could not be related to peer bullying or school bullying but instead with bullying inside the households during the pandemic? If this is so, to what extent these could be due to an important methodological problem that was not controlled or that escaped out of control of researchers?

Response: We appreciate it very much for this good suggestion, and we have done it according to your ideas. Explanations have been added to the discussion as to why the results of other studies are contrary to the results of this study. Since this article only discusses traditional school bullying, it has also been revised in the article, and no specific investigation of cyberbullying has been conducted. You are quite right that family factors will also greatly influence the incidence of school bullying, but this study has not studied this influencing factor, so it is also added to the limitation. Thank you for your comments on my manuscript. We benefited a lot from it. We wish you success in work and good luck!

Reviewer 3 Report

School bullying before and during the pandemic is a serious problem of the institute of education (this applies to school and higher education). Therefore, the relevance of the research undertaken by the authors does not cause any doubt.

However, there are some points in the article that require correction. 1) It is necessary to clearly define the research hypothesis immediately after the goal at the end of the introductory part of the article; 2) it would be desirable to present the methods in more detail, including indicators of internal consistency of the scales used (for example, Cronbach's Alpha); 3) the principle of attributing indicators (included in Table 1) to the presence/absence of victimization of a student based on the use of questionnaires is not clear; 4) it may make sense to compare average indicators at the level of such samples using any criteria? Maybe the "norm" of indicators has changed over time; 5) the results of regression analysis with differentiation of types of bullying also need to be discussed; 6) by design: it would be desirable to disclose all abbreviations in the text (once either in the methods section, or as they occur; 7) since there are gender differences in bullying, it would be desirable to differentiate the results by sex/gender principle as well. I wish the authors success!

Author Response

Dear editor,

Thank you for editor’ and reviewers’ opinions. My article number is ijerph-2131360. These comments are very helpful to improve the quality of the manuscript. Now I response the reviewers’ comments with a point by point and highlight the changes in revised manuscript. Full details of files are listed. Thank you again for your valuable comments.

Yours,

sincerely

  1. It is necessary to clearly define the research hypothesis immediately after the goal at the end of the introductory part of the article;

Response: We appreciate it very much for this good suggestion, and we have done it according to your ideas. We have added hypotheses after the introduction at Line 79-80.

  1. It would be desirable to present the methods in more detail, including indicators of internal consistency of the scales used (for example, Cronbach's Alpha);

Response: We appreciate it very much for this good suggestion, and we have done it according to your ideas. We have described each item of each scale and added the corresponding Cronbach's Alpha at Line 136-156.

  1. The principle of attributing indicators (included in Table 1) to the presence/absence of victimization of a student based on the use of questionnaires is not clear;

Response: We appreciate it very much for this good suggestion, and we have done it according to your ideas. The title for Table 1 have been revised. Table 1 shows the prevalence of school overall bullying victimization and perpetration among secondary school students with different demographic characteristics before and during the pandemic. The original expression was not clear, but now it has been modified. "If the score for a dimension is higher than 1, it demonstrates that the student is being bullied or bullying others in that dimension. If the overall score for the four dimensions is higher than 1, the student is being bullied or bullying others at school." This is the standard of bullying victimization and bullying perpetration.

  1. it may make sense to compare average indicators at the level of such samples using any criteria? Maybe the "norm" of indicators has changed over time;

Response: We appreciate it very much for this good suggestion, and we have done it according to your ideas. You are absolutely right that results can be affected over time, and there are many confounding factors. Therefore, this paper corrects the confounding factors as far as possible. This article also adds this to the discussion in limitations.

  1. the results of regression analysis with differentiation of types of bullying also need to be discussed;

Response: We appreciate it very much for this good suggestion, and we have done it according to your ideas. We've talked about all types of bullying at Line 276-281.

  1. by design: it would be desirable to disclose all abbreviations in the text (once either in the methods section, or as they occur;

Response: We appreciate it very much for this good suggestion, and we have done it according to your ideas. We have disclosed all abbreviations before reference.

  1. since there are gender differences in bullying, it would be desirable to differentiate the results by sex/gender principle as well. I wish the authors success!

Response: We appreciate it very much for this good suggestion. You are absolutely right that bullying is related to gender, but gender has been adjusted as a confounding variable in this paper, mainly to discuss the impact of the COVID-19 pandemic on different types of traditional bullying. Thank you for your comments on my manuscript. We benefited a lot from it. At the same time, the above are also my thoughts for you to ask questions, which we think can better let you understand the content of my article.At last, thank you for your blessing and wish you success in your work.

Reviewer 4 Report

Overall, the manuscript needs more detail on the subject matter. The pandemic issue requires further clarification, particularly regarding how China deals with the schooling system during the Covid19 pandemic. This is critical in determining the validity of comparing school bullying prevalence before and during the pandemic. Furthermore, the study's methodology is insufficient. Even though it involved a large number of students, it may not represent entire China, so the study's title should be revised. The text lacks sample size estimation and sampling method. The manner in which the analysis is carried out is perplexing. The authors must describe the dependent and independent variables in the logistic analysis.

How did you get a total sample of 4436 (refer to line 148) based on the results when you only had 2711 respondents involved during the pandemic? It is invalid to combine samples from before and during the pandemic in the analysis. It introduces bias into the measurement, and different students' perceptions may differ in different situations. It is acceptable if the same person was interviewed before and during the pandemic. Besides that, you must clearly state the validity of the tools used in this study.

The discussion is too brief and lacks sufficient scientific input. The study's findings should be reflected in the conclusion and recommendations.

Author Response

Dear editor,

Thank you for editor’ and reviewers’ opinions. My article number is ijerph-2131360. These comments are very helpful to improve the quality of the manuscript. Now I response the reviewers’ comments with a point by point and highlight the changes in revised manuscript. Full details of files are listed. Thank you again for your valuable comments.

Yours,

sincerely

  1. The pandemic issue requires further clarification, particularly regarding how China deals with the schooling system during the Covid19 pandemic. This is critical in determining the validity of comparing school bullying prevalence before and during the pandemic. 

Response: We appreciate it very much for this good suggestion, and we have done it according to your ideas. In order to give readers a better understanding of the situation of middle schools in China during the pandemic, we added to the discussion at Line 245-254.

  1. Furthermore, the study's methodology is insufficient. Even though it involved a large number of students, it may not represent entire China, so the study's title should be revised.

Response: We appreciate it very much for this good suggestion, and we have done it according to your ideas. Several parts of the research methodology have been supplemented as shown in 2.1,2.3,2.5. At the same time, the title has been modified to more accurately express the research topic.

  1. The text lacks sample size estimation and sampling method.

Response: We appreciate it very much for this good suggestion, and we have done it according to your ideas. Sample size estimation methods and sampling methods have been added in parts 2.2 and 2.3.

  1. The manner in which the analysis is carried out is perplexing. The authors must describe the dependent and independent variables in the logistic analysis.

Response: We appreciate it very much for this good suggestion. The method of analysis is learned by reading other literature, including the setting of the model, such as “Yoon HY, Song TJ, Yee J, Park J, Gwak HS. Association between Genetic Polymorphisms and Bleeding in Patients on Direct Oral Anticoagulants. Pharmaceutics. 2022 Sep 7;14(9):1889. doi: 10.3390/pharmaceutics14091889. PMID: 36145636; PMCID: PMC9501033.”. Binary logistic regression permitted adjustments for potential confounders. This study mainly discussed the impact of the COVID-19 pandemic on the incidence of bullying. Factors other than the COVID-19 pandemic were taken as confounding factors, and other influencing factors other than the pandemic were not discussed in this study. Thank you for your comments on my manuscript. We benefited a lot from it. At the same time, the above are also my thoughts for you to ask questions, which we think can better let you understand the content of my article. We wish you success in work and good luck!

  1. How did you get a total sample of 4436 (refer to line 148) based on the results when you only had 2711 respondents involved during the pandemic? It is invalid to combine samples from before and during the pandemic in the analysis.

Response: We appreciate it very much for this good suggestion, and we have done it according to your ideas. We are very sorry for our careless mistake and it was rectified. The analysis was conducted by separate comparisons between pre-pandemic and pandemic periods.

  1. It introduces bias into the measurement, and different students' perceptions may differ in different situations. It is acceptable if the same person was interviewed before and during the pandemic.

Response: We appreciate it very much for this good suggestion, and we have done it according to your ideas. Although they were not the same group of people before and after the pandemic, they were all middle school students in Shantou City. Meanwhile, confounding factors were corrected to reduce the deviation to a certain extent. You are absolutely right. This is also the limitation of this study, so I have added it to the article.

  1. Besides that, you must clearly state the validity of the tools used in this study.

Response: We appreciate it very much for this good suggestion, and we have done it according to your ideas. We have added the Cronbach's Alpha of the scale at Line 136-156. At the same time, each item in the used scale is described in detail.

  1. The discussion is too brief and lacks sufficient scientific input. The study's findings should be reflected in the conclusion and recommendations.

Response: We appreciate it very much for this good suggestion, and we have done it according to your ideas. It's been expanded in the discussion section. As for English writing, I have polished my article before submitting it for the first time. The agency for polishing is WORDVICE. Thank you for your comments on my manuscript. We benefited a lot from it. We wish you success in work and good luck!

Round 2

Reviewer 1 Report

I am very glad that Authors replayed very quick, and improved the manuscript. This makes the study results valuable in peer violence prevention among adolescents in public health emergencies.

Below please find my comments and remarks.

Still few language errors: line 45: mental, in line 78-79 (H1), line 110 should be formulated in the past tense, line 175, line 244, 268.

1.  In Abstract: study sample characteristics is in line 13 and 16. Please combine.

Keywords are missing, such as adolescent, risk behaviors. In title the term: “traditional” is not needed. However since the study consider health risk behaviors, it should be included in the title and/or keywords  (alcohol consumption, smoking, playing violent video games)

2. Hypothesis is not properly and fully formulated, should be more specific reg. whole analysis and results (prevalence, differences).

3. Results. The title of Table 1 is incorrect. Table 1 incudes also risk behaviors. It is not clear why demographic factors and the risk behaviors have been chosen and shown in Table 1 if there is no continuation in results, for example showing the differences in the prevalence in “pre-pandemic group” and “pandemic group”. Please analyze and show it. Table 1 should be corrected and changed. The comments about this should be include in the text.

4. In Discussion first sentence does not refer to the second one, it is not logical. Line 233 please clarify what study. In line 255 to refer to (r=0.46) is not clear, is incorrect.

The discussion about differences in demographics and risk behaviors between “pre-pandemic group” and “pandemic group”  is missing. When you provide the analysis, please complete.

Line 258-275: please combine the references and write in the end of the paragraph according to the IJERPH/MDPI instructions for authors.

Line 291-306: the comments about cyberbullying and household bullying are pointless, unnecessary, since the study does not address this issues, does not apply to these phenomena.

Line 317-319 is not clear and should be rewritten. THE RESULTS OF THIS STUDY SHOWED THAT THE RECURRENCE OF EPIDEMICS AND FREQUENT EMERGENCIES SUCH AS  BLOCKED ACCESS MIGHT IMPACT ON INCREASE OF SCHOOL BULLYING AMONG ADOLESCENTS.

Author Response

Dear reviewer,

Thank you for editor’ and reviewers’ opinions, and thank you for your affirmation of my revised manuscript. I will continue to work hard. Your comments are very helpful to improve the quality of the manuscript. Now I response the reviewers’ comments with a point by point and highlight the changes in revised manuscript. Full details of files are listed. Thank you again for your valuable comments.

Yours,

sincerely

  1. Still few language errors: line 45: mental, in line 78-79 (H1), line 110 should be formulated in the past tense, line 175, line 244, 268.

Response: We appreciate it very much for this good suggestion, and we have done it according to your ideas. I'm sorry for my our careless mistake. We have corrected the misspelled words and asked my native English speaker colleague to correct the sentences with grammatical problems.

  1. In Abstract: study sample characteristics is in line 13 and 16. Please combine.Keywords are missing, such as adolescent, risk behaviors. In title the term: “traditional” is not needed. However since the study consider health risk behaviors, it should be included in the title and/or keywords  (alcohol consumption, smoking, playing violent video games)

Response: We appreciate it very much for this good suggestion, and we have done it according to your ideas. We have combined the sentences related to the study sample characteristics, added some key words and deleted “traditional” in title.

  1. Hypothesis is not properly and fully formulated, should be more specific reg. whole analysis and results (prevalence, differences).

Response: We appreciate it very much for this good suggestion, and we have done it according to your ideas. I have changed it to "The prevalence of traditional school bullying was different before and during the pandemic".

  1. The title of Table 1 is incorrect. Table 1 incudes also risk behaviors. It is not clear why demographic factors and the risk behaviors have been chosen and shown in Table 1 if there is no continuation in results, for example showing the differences in the prevalence in “pre-pandemic group” and “pandemic group”. Please analyze and show it. Table 1 should be corrected and changed. The comments about this should be include in the text.

Response: We appreciate it very much for this good suggestion, and we have done it according to your ideas. I have changed the title of Table 1 to "Comparison of the prevalence of overall bullying victimization and bullying perpetration by different variables before and during the pandemic" and modified the relevant sections.

  1. In Discussion first sentence does not refer to the second one, it is not logical. Line 233 please clarify what study. In line 255 to refer to (r=0.46) is not clear, is incorrect.The discussion about differences in demographics and risk behaviors between “pre-pandemic group” and “pandemic group” is missing. When you provide the analysis, please complete.Line 258-275: please combine the references and write in the end of the paragraph according to the IJERPH/MDPI instructions for authors.Line 291-306: the comments about cyberbullying and household bullying are pointless, unnecessary, since the study does not address this issues, does not apply to these phenomena.Line 317-319 is not clear and should be rewritten. THE RESULTS OF THIS STUDY SHOWED THAT THE RECURRENCE OF EPIDEMICS AND FREQUENT EMERGENCIES SUCH AS  BLOCKED ACCESS MIGHT IMPACT ON INCREASE OF SCHOOL BULLYING AMONG ADOLESCENTS.

Response: We appreciate it very much for this good suggestion, and we have done it according to your ideas. I have corrected the logical problem between the first and second sentences in the discussion and deleted “(r=0.46)”. Demographic characteristics and risk behavior factors were not explored in the discussion, as the purpose of this study was to examine whether the pandemic caused changes in the prevalence of bullying. Therefore, this part is not discussed in detail.

By reading the notes to authors and other articles in this journal, I found that references are not quoted at the end of paragraphs, but inserted at the end of sentences. Cyberbullying and bullying within the family are cited because some scholars have suggested that these two types of bullying may also be affected during the pandemic. This is a limitation of the article and it needs to be explained why these two types of bullying have not been discussed. At the same time, considering your opinion, I have cut some sentences. Finally, I have revised the sentences in Line 317-319.

Thank you for your comments on my manuscript. We benefited a lot from it. At the same time, the above are also my thoughts for you to ask questions, which we think can better let you understand the content of my article. We wish you success in work and good luck!

Reviewer 2 Report

I think the manuscript has been improved to a good extent, so it could be published in the journal.

Author Response

Dear reviewer,

Thank you for the advice you gave me before, which gave me a lot of inspiration. Finally, thank you for your affirmation of my revised draft. I will continue to work hard in the future. I wish you all the best.

Yours,

sincerely

Reviewer 4 Report

I reviewed a revised version of the manuscript and am pleased with the amount of correction that was made. I only have a few minor concerns to address:

 Methodology - Please provide a brief description of the school policy in place during the Covid19 pandemic and the statistical analysis used in the study, particularly in determining the incidence of bullying before and during the pandemic.

Result - Table 1 - Please indicate the number of samples based on their time difference example before covid19 (n = 3071), delete (N=5782), and because the labeling of the figures is currently ambiguous, please show readers your figures based on the time frame analysis - before and during the pandemic. The analysis to determine the incidence of school bullying should involve samples from before and during the pandemic separately.  It is recommended that you analyze the items from each questionnaire as well as before and during the pandemic, and the results may help you with your discussion.

You are encouraged to seek advice from a professional English proofreader in order to improve the readability of your article.

Author Response

Dear reviewer,

Thank you for editor’ and reviewers’ opinions, and thank you for your affirmation of my revised manuscript. I will continue to work hard. Your comments are very helpful to improve the quality of the manuscript. Now I response the reviewers’ comments with a point by point and highlight the changes in revised manuscript. Full details of files are listed. Thank you again for your valuable comments.

Yours,

sincerely

  1. Methodology - Please provide a brief description of the school policy in place during the Covid19 pandemic and the statistical analysis used in the study, particularly in determining the incidence of bullying before and during the pandemic.

Response: We appreciate it very much for this good suggestion. Since the school did not disclose the management regulations during the pandemic, I could only obtain some information based on interviews with students and teachers. It is already under discussion. For example: “schools had adopted tightened infection prevention and control. Students were not allowed to leave school at will, only for serious reasons, and students were required to undergo frequent nucleic acid tests. In order to ease students' rebellious mood, the school organized more learning and social activities such as learning lectures and fun sports games”. We added something to the statistical analysis. However, as for the specific statistical analysis method of model setting, I explained in detail in 2.4.

  1. Result - Table 1 - Please indicate the number of samples based on their time difference example before covid19 (n = 3071), delete (N=5782), and because the labeling of the figures is currently ambiguous, please show readers your figures based on the time frame analysis - before and during the pandemic. The analysis to determine the incidence of school bullying should involve samples from before and during the pandemic separately.  It is recommended that you analyze the items from each questionnaire as well as before and during the pandemic, and the results may help you with your discussion.You are encouraged to seek advice from a professional English proofreader in order to improve the readability of your article.

Response: We appreciate it very much for this good suggestion. Your comments have inspired me a lot. I have deleted "(N=5782)" in Table 1. Because this is a separate study before and during a pandemic, the two samples should not be combined. According to your comments, We have added the corresponding sample size before and during the pandemic at Results. Thank you for your comments on my manuscript. We benefited a lot from it. We wish you success in work and good luck!